# Prokaryote Distribution Patterns along a Dissolved Oxygen Gradient Section in the Tropical Pacific Ocean

**DOI:** 10.3390/microorganisms11092172

**Published:** 2023-08-28

**Authors:** Peiqing He, Huan Wang, Jie Shi, Ming Xin, Weimin Wang, Linping Xie, Qinsheng Wei, Mu Huang, Xuefa Shi, Yaqin Fan, Hao Chen

**Affiliations:** 1Key Laboratory of Science and Technology for Marine Ecology and Environment, First Institute of Oceanography, Ministry of Natural Resources, 6 Xianxialing Road, Qingdao 266061, China; hepeiqing@fio.org.cn (P.H.); wanghuan@fio.org.cn (H.W.); shijie@fio.org.cn (J.S.); xinming@fio.org.cn (M.X.); xielinping@fio.org.cn (L.X.); weiqinsheng@fio.org.cn (Q.W.); fayaqin@fio.org.cn (Y.F.); 2Laboratory for Marine Biology and Biotechnology, Pilot National Laboratory for Marine Science and Technology, 168 Wenhai Middle Road, Aoshanwei, Jimo District, Qingdao 266071, China; 3Center for Ocean and Climate Research, First Institute of Oceanography, Ministry of Natural Resources, 6 Xianxialing Road, Qingdao 266061, China; wwmin@126.com; 4Key Laboratory of State Oceanic Administration for Marine Sedimentology & Environmental Geology, Ministry of Natural Resources, 6 Xianxialing Road, Qingdao 266061, China; huangmu@fio.org.cn (M.H.); shixuefa@fio.org.cn (X.S.)

**Keywords:** dissolved oxygen, Thaumarchaeota, Nitrospinae, chlorophyll *a*, nitrate, the Tropical Pacific Ocean

## Abstract

Oceanic oxygen levels are decreasing significantly in response to global climate change; however, the microbial diversity and ecological functional responses to dissolved oxygen (DO) in the open ocean are largely unknown. Here, we present prokaryotic distribution coupled with physical and biogeochemical variables and DO gradients from the surface to near the bottom of a water column along an approximately 12,000-km transect from 13° N to 18° S in the Tropical Pacific Ocean. Nitrate (11.42%), temperature (10.90%), pH (10.91%), silicate (9.34%), phosphate (4.25%), chlorophyll *a* (3.66%), DO (3.50%), and salinity (3.48%) significantly explained the microbial community variations in the studied area. A distinct microbial community composition broadly corresponding to the water masses formed vertically. Additionally, distinct ecotypes of Thaumarchaeota and Nitrospinae belonging to diverse phylogenetic clades that coincided with specific vertical niches were observed. Moreover, the correlation analysis revealed large-scale natural feedback in which chlorophyll *a* (organic matter) promoted Thaumarchaeotal biomass at depths that subsequently coupled with *Nitrospina*, produced and replenished nitrate for phytoplankton productivity at the surface. Low DO also favored Thaumarchaeota growth and fueled nitrate production.

## 1. Introduction

The global ocean is a dynamic ecosystem. In the euphotic zone of the global ocean, biological matter is synthetized through oxygenic photosynthesis. When organic matter is exported downward and remineralized at depth, the dissolved oxygen (DO) is consumed [1]. A significant decrease in oxygen concentration typically occurs in midwater depths [2]. In some areas of the open ocean, high rates of phytoplankton productivity coupled with sluggish ventilation can reduce oxygen concentrations up to <20 μmol L^−1^. This leads to the formation of oxygen minimum zones (OMZs), which are expected to expand and intensify in the future [2,3].

When DO is consumed, shifts in the microbial composition might occur due to the supply of alternative electron acceptors, such as nitrate and sulfate. Such shifts are obvious in OMZs. In the Eastern Tropical North Pacific Ocean (ETNP), deoxygenation is found to alter microbial richness, with 5% of microbial operational taxonomic units (OTUs) being changed in response to each 1 μmol L^−1^ change in DO [4]. Some OTUs may increase with deoxygenation as others decrease; moreover, numerous OTUs may have expanded or contracted ranges [4]. Generally, typical bacteria such as SAR324, ARCTIC96BD-19/SUP05, and *Nitrospina* are abundant in OMZs [5]. However, shifts in OTUs may enhance the loss of fixed nitrogen and can even fuel H_2_S production [6,7]. Consequently, changes in local and global biological production and nutrient cycles have occurred in response to changing DO levels in the ocean [5].

DO can also play an important role in determining oceanic nitrification, which is a two-step process catalyzed by ammonia-oxidizing and nitrite-oxidizing microorganisms. Ammonia-oxidizing archaea (Thaumarchaeota), which are among the most widespread and abundant microorganisms in the ocean, are particularly abundant in mesopelagic zones [8] and in OMZs [9,10,11]. Nitrospinae, which are abundant in some parts of the dark ocean, are major nitrite-oxidizing bacteria in OMZs [12,13]. High levels of nitrification primarily occur at the base of the euphotic zone, where the DO typically decreases via the remineralization of sinking organic matter [14]. In general, ammonia oxidation rates are higher than nitrite oxidation rates in the epipelagic zone, while the opposite is true below the epipelagic zone in the South China Sea and Western Pacific [15], North Atlantic Ocean [16,17], and Southern California Bight [14,18]. In the OMZs of the Eastern Tropical South Pacific (ETSP), the subsurface nitrite oxidation maximum in the upper oxycline was found at deeper levels than the ammonia oxidation maximum [11,16]. Moreover, both ammonium oxidation rates and nitrite oxidation rates dropped sharply at DO ≤ 1 μmol L^−1^ in these OMZs [19,20,21,22]. However, both processes were still detectable at nanomolar oxygen concentrations (5–30 nmol L^−1^), although there was different oxygen dependence between ammonium and nitrite oxidation [23]. As highlighted above, the majority of studies on changes in microbial communities in response to DO changes conducted to date have been confined to OMZs.

The tropical Pacific is becoming substantially deoxidized in response to global warming. There is an OMZ in the Eastern Tropical South Pacific Ocean (ETSP) and the Eastern Tropical North Pacific Ocean (ETNP), where high rates of phytoplankton productivity coupled with sluggish ventilation can deplete the oxygen concentrations to below 20 μmol L^−1^ [2,3]. Fifty-year oxygen records from the Eastern Tropical Pacific Ocean have revealed expanding OMZs, with declines of 0.09–0.34 μmol L^−1^ y^−1^ occurring at 300–700 m [24]. Therefore, it is essential to understand the interplay between DO shifts and nitrogen cycles mediated by microbial community diversity and dynamics in the open ocean. In this study, we investigated large-scale geographic patterns of microbial diversity across DO gradients in the water column of the Eastern Tropical Pacific Ocean based on an analysis of 16S rRNA amplicons. We also explored the interplay between the nitrate concentration, chlorophyll *a* (photosynthetic primary production), DO level, and nitrifying taxa (Thaumarchaeota and *Nitrospina*) to reveal the process responsible for regulation of the nitrogen cycle. The results of this study will facilitate the development of a predictive understanding of future oceanic nitrogen cycles under global climate change.

## 2. Materials and Methods

### 2.1. Station Locations and Sampling Process

Seawater samples were collected from the Tropic Pacific Ocean aboard the Research Vessel (R/V) Xiang 1 (March to May 2019) during a global cruise. The cruises were organized by the First Institute of Oceanography, Ministry of Natural Resources of China (MNR). The sampling stations were located in the Mariana Basin (Station V2114) and extended southeastward. The studied section crosses the equator near 163° W (Station V1713), reaches the southern end of the Northeast Pacific Basin (Station V1312 and V0605), then extends to the Bauer Basin and the Youpanqui Basin (Station IV1713 and IV2016) (Figure 1).

At each station, 19–21 depths were sampled from 10 m below the surface to near the bottom of the water column (Appendix A) using Niskin bottles attached to a rosette equipped with Sea-Bird SBE 911 plus conductivity–temperature–depth (CTD) sensors (Sea-Bird Electronics Inc., Bellevue, WA, USA). At each depth, 2 L of seawater was filtered through 0.45-µm cellulose-acetate membranes (Whatman, Florham Park, NJ, USA); after which, the filtered membranes (10–300 m deep) were stored at −80 °C until the subsequent chlorophyll *a* analyses. The 200-mL filtrates were stored in polyethylene vials at −20 °C for subsequent nutrient analyses, including for nitrate (NO_3_^−^), silicate (SiO_3_^2−^), and phosphate (PO_4_^3−^). At each depth, 5 L of collected water was filtered through 0.22-µm acetate membranes using a peristaltic pump. The filter membranes were then stored at −80 °C until the subsequent DNA extraction.

### 2.2. Physical and Chemical Analyses

Water temperature and salinity were determined with onboard SED 911 plus CTD sensors. Water dissolved oxygen (DO) was measured onboard using the iodometric method [25]. In addition, a DZS-706 multiparameter analyzer (INESA Scientific Instrument Co., Ltd., Shanghai, China) was used onboard to determine the pH values, which were corrected to values at the in situ temperature and pressure conditions based on the Chinese national standard of GB/T 12763.4-2007 [26]. Nutrient (NO_3_^−^, SiO_3_^2−^, and PO_4_^3−^) analyses were conducted at the laboratory after the cruise using colorimetric methods and a QuAAtro nutrient automatic analyzer (Bran+Luebbe Gmbh, Germany). NO_3_^−^ concentrations were measured using the standard pink azo dye method [27], while PO_4_^3−^ concentrations were measured with the molybdenum blue method [28,29], and SiO_3_^2−^ concentrations were measured using the molybdosilicic acid method [30]. The chlorophyll *a* concentration was determined as described by Parsons et al. [31] using a TD-700 Laboratory Fluorometer (Turner Designs, San Jose, CA, USA).

### 2.3. DNA Extraction, 16S rRNA Polymerase Chain Reaction (PCR) Amplification, and Sequencing

The total genomic DNA was extracted from each sample using a FastDNA Spin Kit for Soil (MP Biomedicals, Santa Ana, CA, USA) following the manufacturer’s instructions. The V3–V4 hypervariable regions of 16S ribosomal RNA genes were then PCR-amplified using the universal primers 341F (5′-CCTAYGGGRBGCASCAG-3′) and 806R (5′-GGACTACNNGGGTATCTAAT-3′). During amplification, a barcode unique to each sample was added to each sequence. Each 30-µL PCR reaction contained 15 μL of Phusion^®^ High-Fidelity PCR Master Mix (New England Biolabs, Beijing, China), 0.2 μL of each primer (5 μmol L^−1^), and about 10 ng of template DNA. The amplification cycling program comprised an initial denaturation at 98 °C for 1 min, followed by 30 cycles of denaturation at 98 °C for 10 s, annealing at 50 °C for 30 s, and extension at 72 °C for 30 s, with a final extension at 72 °C for 5 min. The PCR products were then purified using the GeneJET^TM^ Gel Extraction Kit (Thermo Fisher Scientific, Waltham, MA, USA). Sequencing libraries were generated using an Ion Plus Fragment Library Kit 48 rxns (Thermo Fisher Scientific, Waltham, MA, USA). Libraries were sequenced on an Ion S5^TM^ XL platform, and 400-bp/600-bp single-end reads were generated.

### 2.4. Bioinformatics Analysis and Statistical Analysis

Single-end reads were assigned to samples based on their unique barcodes; after which, the barcodes and primer sequences were removed. Quality filtering of the raw reads was performed to obtain high-quality clean reads [32] (V1.9.1, http://cutadapt.readthedocs.io/en/stable/, accessed on 3 February 2021). The reads were then compared with the Silva reference database (https://www.arb-silva.de/, accessed on 3 February 2021) (Quast et al., 2013) using the UCHIME algorithm (UCHIME algorithm, http://www.drive5.com/usearch/manual/uchime_algo.html, accessed on 3 February 2021) [33] to detect and remove chimera sequences [34], resulting in the final effective read dataset. A further sequence analysis was conducted using the Uparse software program (v.7.0.1001, http://drive5.com/uparse/, accessed on 3 February 2021). Sequences with ≥97% nucleotide similarity were assigned to the same OTUs. A representative sequence from each OTU was used for further taxonomic annotation. Taxonomic information was assigned to each representative sequence by comparison to the Silva database (https://www.arb-silva.de/, accessed on 3 February 2021) using the Mothur classification algorithm [35]. OTU abundance information was normalized based on the number of sequences in the sample with the least number of sequences.

The QIIME platform (Version 1.7.0) was used to compute Alpha diversity indices, including the Shannon Index and ACE Index. Hierarchical clustering was used to identify depth-based water masses and community structures in the “vegan” R package (v. 4.1.1) [36]. For water masses, the temperature and salinity values were first standardized to *Z*-score values (means of 0 and one unit SD). Next, Euclidean distances coupled with the Ward linkage method were calculated. For the community structure, Hellinger-transformed OTU abundances coupled with the Ward linkage method were calculated. Structural variations were partitioned using a permutational multivariate analysis of variance (PERMANOVA) from the “vegan” R package. Analyses of similarities (ANOSIM) were performed from the “vegan” R package. A redundancy analysis (RDA) was conducted using a Hellinger distance matrix from the “vegan” R package to assess environmental controls on microbial community variations. The physical, chemical, and biotic variables were standardized to *Z*-score values. The environmental parameter measurements included temperature; salinity; pH; and concentrations of NO_3_^−^, SiO_3_^2−^, PO_4_^3−^, DO, and chlorophyll *a*. The linear discriminant analysis effect size (LEfSe) was used to identify bacteria indictive of different community structures [37]. Metabolic function prediction of the microbial communities was performed using PICRUSt (Phylogenetic Investigation of Communities by Reconstruction of Unobserved States) [38]. Spearman correlations between species abundances and dissolved oxygen (DO) were calculated using the Statistical Package for Social Sciences (SPSS) (v. 22.0) software program (IBM Corp., Armonk, NY, USA), with *p* < 0.01 set as the threshold for statistical significance and |*R*| ≥ 0.5 indicating that two variables had a moderate-to-strong relationship [39]. 

Phylogenetic trees were constructed using the neighbor-joining (NJ) algorithm, as implemented in the MEGA11 software program (v. 11.0.11) [40]. Bootstrapping of the phylogenetic reconstructions was performed by resampling 1000 times, with bootstrap values < 50% not shown in the visualizations.

### 2.5. Primers for Quantitative PCR (qPCR)

Bacterial 16S rRNA, Thaumarchaeotal 16S rRNA and ammonia monooxygenase subunit A (*amoA*), and *Nitrospina* 16S rRNA genes were quantified using the primers shown in Table 1.

PCR amplifications were conducted using 1 μL of TransStart Taq DNA polymerase (2.5 U, TransGen Biotech, Beijing, China), 2 μL of input DNA, 1 μL of each primer (0.2 μmol L^−1^ final concentration), 5 μL of buffer, 4 μL of dNTPs, and ultra-pure sterile water added to a final volume of 50 μL. The PCR program consisted of an initial denaturation at 94 °C for 5 min, followed by 35 cycles of denaturation at 95 °C for 10 s, annealing at temperatures specified for primers (Table 1) for 10 s, and extension at 72.0 °C for 30 s. A final extension at 72 °C for 10 min was conducted after the amplification cycles. The PCR products were then visualized using agarose gel electrophoresis. Products of successful amplifications were subsequently purified using a purification kit (TianGen Biotech, Beijing, China) and cloned into pEASY-T1 cloning vectors (TransGen Biotech, Beijing, China) according to the manufacturer’s instructions. At least 10 positive transformants were randomly selected for each primer set and sequenced using the M13F and M13R primers. Sequences were evaluated through Basic Local Alignment Search Tool (BLAST) searches of the National Center for Biotechnology Information (NCBI) database to validate target genes.

### 2.6. qPCR Analysis

qPCR assays were conducted on a LightCycler 96 (Roche Inc., Branchburg, NJ, USA) equipped with LightCycler Application software (v. 1.1). The qPCR reaction mixtures included 10 μL of SYBR Premix Ex Taq II (Takara Bio, Dalian, China), 2 μL of DNA, 1 μL of each primer (to a final concentration of 0.02 μmol L^−1^), and ultra-pure sterile water added to a final volume of 20 μL. qPCR reactions were conducted in 0.1 mL eight-strip qPCR tubes. The qPCR cycle consisted of an initial denaturation at 94 °C for 30 s, followed by 40 amplification cycles of denaturation at 94 °C for 5 s, annealing at temperatures specific for each primer (Table 1) for 30 s, and extension at 72.0 °C for 30 s. Following amplification, a melt curve analysis was conducted at 95 °C for 15 s, then at 65 °C for 15 s and at 95 °C for 15 s, to rule out nonspecific PCR amplification products.

### 2.7. Preparation of qPCR Standards

Amplification products of bacterial 16S rRNA, Thaumarchaeotal 16S rRNA and *amoA*, and *Nitrospina* 16S rRNA genes were evaluated by agarose gel (1%) electrophoresis; after which, the bands were isolated and purified using a purification kit (TianGen Biotech, Beijing, China). Isolated bands were then used as qPCR standards. Purified products were quantified using Picogreen dsDNA Quantitation Reagent (Solarbio Science & Technology Co., Beijing, China), with a known concentration of calf thymus DNA (1 mg mL^−1^) as the quantification standard. Gene copy numbers were calculated as described by Stubner [45]. To obtain log-linear relationships between gene copies and cycle thresholds (*Cts*), the standards genes were serially diluted between 1 × 10^1^ and 1 × 10^9^ copies mL^−1^; after which, the dilutions were used to construct standard curves for different genes. All analyses were performed with the LightCycler Application software program (v. 1.1). Correlation coefficients (*R*^2^ values) for the standard curves ranged from 0.990 to 0.999, with slope values ranging from −3.10 to −3.58, providing estimated amplification efficiencies of 90–110%.

Environmental DNA extracts were evaluated for inhibitory effects before quantification by determining *Ct* values for a dilution series of DNA [45]. PicoGreen assays revealed that template dilutions of approximately 10 ng μL^−1^ resulted in the lowest *Ct* values; therefore, this concentration was chosen for all qPCR assays. Reactions were performed in duplicate for the standards and in triplicate for the samples, with average values reported for each sample.

## 3. Results and Discussion

### 3.1. Temperature and Salinity Profiles

After standardization of the temperature and salinity values to *Z*-scores, all samples fell into six groups during the base of cluster analysis (Table 2). The temperature and salinity profiles are shown in Figure 2a,b, Figure 3a, and Appendix A. The surface water (SW) comprised SW1, SW2, and SW3, primarily ranging from depths of 10–150 m. The lower boundary of SW was located at the base of the chlorophyll *a* maximum. Vertical uniformity of the temperature (10–50 m and 10–100 m) and salinity (10–30 m and 10–50 m) was also observed. The surface temperatures ranged from 23.31 to 27.65 °C between 18° S and 13° N. These values generally declined with increasing latitude, with V1312 being slightly lower than that of V0605. The salinity decreased from 36.17 at 18° S to 35.07 at the equator station (V1713) and then to 34.70 at V2114. Subsurface water (SSW) was primarily distributed from 200- or 300- to 500-m depths, with the upper boundary reaching up to 150 m at V1312. SSW displayed a rapid decrease in temperature and salinity. Colder waters at a depth of ~150 m at V1312 and IV1713 showed a clear signal indicating upwelling. Intermediate layer water (IW) was primarily distributed between depths of 800 and 2000 m, the temperature decreased gradually from 6.09 to 2.22 °C. Below the SW, the salinity gradually decreased to a minimum of 34.49–34.52. The salinity minimum was typically reached at ~800-m depths, with 1000 m occurring at V1312 and 300 m at V2114. The deep (bottom) layer (DW) was distributed below 2000 m, with the maximum thickness reaching up to ~4000 m. The temperatures (1.25–1.88 °C) and salinities (34.64–34.70) within this large volume were quite uniform.

### 3.2. Dissolved Oxygen (DO) Profile

As shown in Figure 2c and Figure 4a, DO in the surface water decreased from 232.09 and 245.49 µmol L^−1^ at high latitudes (IV1713 and IV2016) to 170.15 µmol L^−1^ at the equator station (V1713), then slightly increased at V2114 (176.55 µmol L^−1^). A DO mixing layer with a thickness of 50–200 m occupied the surface layer, while a much thinner DO mixed layer was found at V1312. The DO value dropped rapidly from depths of 50 to 200 m, with the minimum values being observed at 300 and 500 m. The DO minimum was found at a deeper depth (500 m) at the equator and 13° N stations. The DO increased gradually from the minimum value with the depth. The DO minimum values decreased from 18° S (V2016, 66.86 µmol L^−1^) to 5° S (V1312, 14.30 µmol L^−1^), then increased to 13° N (V2114, 61.67 µmol L^−1^). However, the value at IV1713 (26.04 µmol L^−1^) was slightly lower than at V0605 (30.99 µmol L^−1^). Lower DO minimum values of 14.30 and 26.04 μmol L^−1^ were measured in the upwelling stations. The DO of 14.30 µmol L^−1^ observed near 5° S (V1312) was slightly lower than that of 20 µmol L^−1^, which is the level used to define an OMZ [46].

### 3.3. Nutrients and Chlorophyll a

The concentrations of nitrate (NO_3_^−^), silicate (SiO_3_^2−^), and phosphate (PO_4_^3−^) were low at ocean surfaces but increased notably at the depth range in which the temperature decreased; after which, they accumulated with the water depth (Figure 4). The mean concentrations of NO_3_^−^, SiO_3_^2−^, and PO_4_^3−^ of the full-depth profiles showed a similar latitudinal trend, increasing from 18° S to the equator, then declining at 13° N. Chlorophyll *a* was measured at depths ≤ 150 m, with peak values observed at 75–150 m. The mean concentrations of chlorophyll *a* broadly increased from 18° S to the equator, then declined at 13° N; however, a lower chlorophyll *a* level was found at V0605 than IV1713 (Figure 2d and Figure 5).

### 3.4. Patterns in Prokaryotic Diversity

The ACE and Shannon Indexes displayed bell-shaped patterns along the water depth and DO gradients (Figure 3c,d). As shown in Appendix A, in the upper layer, the ACE Index was the lowest at 10 m, and it peaked at the base of the chlorophyll *a* layer at V2114, V1312, IV1713, and V0605 at a depth of 200 m, while the peak values at 200 and 300 m were comparable at V1713 and IV2016. At the DO minimum depths, the ACE Index showed a slight decrease at V2114, V1312, IV1713, and V0605, then increased with the depth. The ACE value peaked again at ~150 μmol L^−1^ DO (2000–3000 m), then decreased with the increasing depth. This observation was similar to the results reported by Beman and Carolan [4], who found that bacterial richness reached higher values at the base of the euphotic zone and decreased within the OMZ. The low DO might reduce the richness due to the varying oxygen tolerances of microorganisms, as suggested by Bryant et al. [47]. The Shannon Index was also the lowest at 10 m, then increased at the DO minimum depths in V2016, V1312, IV1713, and V2114. Thus, at the DO minimum values depths in V1312, IV1713, and V2114, the ACE Index decreased as the Shannon Index increased. These findings suggested that some rare OTUs might be contracted as DO decreases, which is in accordance with the results reported by Bryant et al. [47]. The Shannon Index peaked at ~3500 m, then declined with the depth.

### 3.5. OTUs Change with Dissolved Oxygen

The Pearson correlation analysis revealed that 116 OTUs shifted in abundance in response to changes in the DO (*p* = 0.01; |R| ≥ 0.5). As shown in Table 3, 82 OTUs were significantly negatively correlated with DO, suggesting that these OTUs may increase with deoxygenation. These OTUs belonged to Thaumarchaeota, Euryarchaeota, Marinimicrobia (formerly known as SAR406), Acidobacteria, Verrucomicrobia, Nitrospinae, Deltaproteobacteria (e.g., SAR324), Gemmatimonades, and Actinobacteria (e.g., organic sulfur transformer Acidimicrobiia). Most Alphaproteobacteria OTUs were significantly positively correlated with DO, indicating that they decreased with deoxygenation. Significant negative correlations between most Gammaproteobacteria OTUs and DO were also observed, such as SUP05, which may detoxify sulfides using nitrates as electronic acceptors [48]. Generally, “typical” sulfur oxidizers (e.g., SAR11, AR324, and SAR406) and nitrite oxidizers (e.g., *Nitrospina*) are abundant in OMZs [5,22]. Therefore, our results highlight the fact that decreases in DO in the open ocean might also have impacted the abundance patterns of microbial taxa that are tightly coupled to N and S cycling.

### 3.6. Community Composition and Structure

The redundancy analysis (RDA) demonstrated that community variation was significantly related to nitrate (11.42%), temperature (10.90%), pH (10.91%), silicate (9.34%), phosphate (4.25%), chlorophyll *a* (3.66%), DO (3.50%), and salinity (3.48%). No significant differences in community composition (OTUs) were observed among V1713, V1312, and V0605 or between IV1713 and IV2016 (PERMANOVA and ANOSIM *p* > 0.05), suggesting that geographic distance influenced microbial assemblages. The hierarchical clustering analysis revealed five clusters, suggesting there were five distinct community structures along the water column, broadly coinciding with the water mass of our study system (Figure 3a,b). These findings were consistent with those of previous studies that showed water masses can act as ecological boundaries [49].

With the exception of Station V0605 at 30 m, the depth from the surface to 75–200 m formed one cluster, and the low boundary was at the chlorophyll *a* maximum depth or the base of the chlorophyll *a* maximum. The prokaryotic assemblage found in this region was referred to as the upper surface community. At this depth range, the samples of each station formed a subbranch, indicating that vertical mixing could control the microbial composition at the upper surface. The indicator taxa *Candidatus* pelagibacter ubique in the base of the LEfSe analysis (Threshold 4) comprised 3–25% of the total classified reads in the upper surface community. *Candidatus* pelagibacter ubique belongs to the SAR11 clade and is the most successful group in the upper surface of ultraoligotrophic oceans [50]. In addition, the most abundant Cyanobacteria were observed at ≤150 m, including unknown cyanobacteria at V1713, *Synechococcals* (up to 25%) at V2114, and *Synechococcals* and *Prochlorococcus* at IV2016. These findings are in accordance with the results of other studies from this area [51,52,53].

With the exception of V2114, the depths ranging from 10–150 m to 150–300 m, as well as those of 3500 and 5000 m at V1713, formed one cluster that represented the surface community. In the surface community cluster, the indicator taxa *Alteromonas* comprised 18–70% of the total population. *Alteromonas* species can colonize large particles and degrade a wide range of organic substrates [54]. Additionally, *Sulfitobacter* comprised 22–44% of the microbial community at IV1713 in this cluster. *Sulfitobacter* species have been shown to exert sulfite-oxidizing and the degrading of algae-derived dimethyl sulfoniopropionate (DMSP) [55].

Depths from 200–500 m to 800–1000 m of the majority stations formed one cluster that represented the subsurface community, with the low boundary of V2114 at 500 m. Depths at 800–1000 m had the minimum salinity, suggesting that salinity might play a crucial role in limiting microbial diffusion. Thaumarchaeota, which is indictive of ammonia oxidizing, comprised 7–48% of the community in this cluster. This finding was in line with the results of a previous study that showed Thaumarchaeota was among the most abundant taxa in the subsurface community [8]. The indicator taxa also included Acidimicrobiia [56], SUP05, Deltaproteobacteria, and Thermoplasta. Prevalent chemolithotrophic activities, including ammonia oxidation, sulfur oxidation, and CO_2_ fixation, have also been reported in the mesopelagic zone of the North Pacific Subtropical Gyre (NPSG) [57,58,59].

Depths from 800–1500 m to 2500 and 3000 m formed one cluster that represented the intermediate community, with the low boundary at V2114 reaching 1250 m. In line with the results of the functional prediction, this cluster contained common indicator species of the deep community, including *Marinobacter*, Alcanivoracaceae, and *Altererythrobacter* (Sphingomonadaceae), which might contribute to the degradation of carbohydrates, hydrocarbons, and lipids [60,61,62]. The methyltrophic bacteria *Methylophaga* [63] and Methylococcales [64], bacterial predator Bradymonadales [65], and putative S-oxidizer SAR324 [66] were also among the most abundant microbial groups in this region.

Depths from 2000 to 3500 m to near the bottom also clustered together, representing the deep community. This community contained high levels of Firmicutes, Bacteroidetes, Tenerictes, Fusobacteria, Acidobacteria, and Chloroflex. These organisms include potential intestinal members, animal parasites, and symbionts that are capable of hydrocarbon degradation and fermentation [67,68,69,70].

### 3.7. Vertical Distribution of Thaumarchaeota and Thaumarchaeota Ecotypes

Our findings suggested that Thaumarchaeota were the most abundant taxa, spanning nearly the entire water column. As shown in Figure 5, the mean relative abundance of Thaumarchaea among different stations was 9–20%, while they showed a peak abundance of up to 48% at V1713 (500 m). Vertically, an obviously high level of Thaumarchaeota was observed below the base of the chlorophyll *a* layer, where DO increased from the minimum value and the nitrate concentrations steadily increased. This depth primarily ranged from 500 to 1500 m, with the upper boundary reaching up to 300 m at V2114. Notably, in the deep part of the DO mixed layer at V1713 and V2114, where nitrate had just begun to increase rapidly, a Thaumarchaeotal peak overlapping the chlorophyll *a* layer was measured. Moreover, a high level of Thaumarchaeota was also found between 2000 and 3000 m, as well as a slight peak near the bottom (4000–5000 m).

A total of 82 Thaumarchaeal OTUs (97% level) were obtained, from which the 28 most abundant OTUs were used to construct a phylogenetic tree (Figure 6a). All OTUs for this tree comprised ≥0.05% of the total population in at least one sample. The first phylogenetic clade comprised seven OTUs that were mainly located at 150–200 m, representing the shallow water ecotype. Most OTUs belonged to the second phylogenetic clade. Within this clade, 18 OTUs showed the maximum relative abundance at 300–500 m, although the total range extended to about 1000–2500 m. These 18 OTUs represented the intermediate water ecotype. Additionally, three OTUs clustering into the second clade displayed the maximum relative abundance at 2500–4500 m or at the bottom of the water column. These OTUs represented the deep (bottom) water ecotype.

Among the shallow water ecotype, OTU 7 was dominant and contributed to the total abundance of Thaumarchaeota in the euphotic zone. OTU 7 was the most abundant OTU at IV2016 (150–200 m), V1713 (75–150 m), and V2114 (150–200 m), which had a corresponding temperature range of 18.17–26.92 °C. OTUs 291, 277, 268, and 207 were the most abundant OTUs at 200 m (18.17°C) at V2114, with a total abundance of 11%. At 200 m at IV1713, the temperature was 13.56 °C, and the total abundance of shallow water ecotypes was 0.5%. These findings indicated that these shallow water ecotypes may thrive in high-temperature surface waters. Consistently, BLAST searches of the NCBI database indicated that OTU 7 was 100% homologous to sequences of *Candidatus* Nitrosopelagicus brevis from the surface of the subtropical North Pacific. Additionally, OTU 207 showed 100% similarity with a clone from the Western Pacific warm pool. OTU 7 also displayed high abundances of 27% and 13% at the chlorophyll *a* maximum layer at V1713 (100 m) and at V2114 (150 m), respectively. The temperature–salinity profile revealed strong mixing in the euphotic zone of V1713 and V2114 (Appendix A). We speculated that vertical mixing caused NO_3_^−^ to be swept up to the euphotic zone, reducing the nutrient substrate competition between photoautotrophic phytoplankton and Thaumarchaeota. This, in turn, drove the overlap of OTU 7 and the chlorophyll *a* layer. However, at 150 m at V0605 (19.65 °C) and IV1713 (17.94 °C), the temperature was equivalent to that at 150 m at IV2016 (20.16 °C), but the total abundance of the shallow water ecotype of Thaumarchaeota was only 0.03%, respectively. The stratification below 150 m at V0605 and IV1713 might limit the upward transport of nutrients. In the nitrogen limited upper layer, phytoplankton excluded the chemoautotrophic nitrifiers that are less efficient at using ammonia for growth. Therefore, the shallow water ecotype was primarily located in the euphotic zone, where both the temperature and nutrient substrate levels were high due to strong mixing. Recent data have also demonstrated an increase in ammonia-oxidizing archaea due to in situ growth at the surface and not to mixing with deeper water masses [71,72].

At least 18 OTUs representing the intermediate water type peaked at 500 m, where they comprised 19–45% of the relative abundance. Moreover, the high abundance of these 18 OTUs could extend to 1000–2500 m. Among these, OTU 13 displayed 100% similarity with sequences from the tropical Mexican Pacific OMZ, while OTU 73 displayed 100% similarity with sequences from the permanent OMZ of the Eastern Tropical South Pacific. Additionally, the abundance of OTU 14, which showed 100% similarity with a sequence from the tropical Mexican Pacific OMZ, reached 15% at 500 m at V1713. Several other OTUs with low relative abundances, such as OTU 2913, displayed the highest similarity of 98.71% with a sequence from the Northeast Subarctic Pacific OMZ (1400 m deep). OTU 7234 displayed the highest similarity of 99.23% with a sequence from the Northeast Subarctic Pacific OMZ (1000 m deep). Thus, the intermediate water ecotype of Thaumarchaeota might be homologous with sequences from OMZs, indicating that these organisms have low oxygen preferences.

Three OTUs were responsible for the total relative abundance of Thaumarchaeota below 2000 m. These OTUs, which formed a subbranch within the intermediate ecotype in the phylogenetic tree, belonged to Nitrosopumilaceae. These OTUs displayed the highest similarity of 98.22–99.23% with sequences from deep seawater, as well as with the hydrothermal plume and surrounding water mass sequences. OTU 55 constituted 5% of the relative abundance between 2500 and 4500 m and displayed the highest relative abundance of 17% at the bottom at IV2016. The relative abundance of OTU 55 also showed a linear positive relationship with OTU 7171 (*R*^2^ = 0.92), and these OTUs comprised the total abundance of Thaumarchaeota below 2000 m.

Overall, our findings indicate the presence of three vertically segregated ecotypes of Thaumarchaeota corresponding to specific phylogenetics with cosmopolitan distribution. This differs from the findings of prior studies of thaumarchaeal ammonia monooxygenase subunit A gene sequences that suggested shallow and deep ecotypes in the ocean [43,73,74]. These results might be accounted for by limited sampling that missed the deep (bottom) ecotype. On the other hand, the deep (bottom) water ecotype and intermediate water ecotype observed in the present study formed a branch within the phylogenetic tree, suggesting a closer phylogenetic relationship.

### 3.8. Vertical Distribution of Nitrospinae and Nitrospinae Ecotypes

Nitrospinae are the most abundant and globally distributed nitrite-oxidizing bacteria in the ocean [75]. As shown in Figure 5, the mean abundance of Nitrospinae between 18° S and 13° N was 0.13–0.24%. Moreover, a peak of Nitrospinae was evident at 150–200 m at IV2016, V0605, V1713, and V2114, and they accounted for up to 0.94% of the total abundance at V1713 at 150 m. Below this depth range, Nitrospinae peaks were found at 1000 m at IV2016 and 800 m at IV1713, which are deeper than the peaks of Thaumarchaeota. However, at 1000 m at V0605 and 500–1000 m at V1312, the peaks overlapped with that of Thaumarchaeota. Moreover, below the base of the peaks, slight increases in Nitrospinae were also found, which was consistent with Thaumarchaeota.

We found a total of 22 OTUs related to Nitrospinae, and 8 OTUs displayed diverse phylogenetic clades corresponding to vertically different niches (Figure 6b). The surface water ecotype (150–200 m), such as OTU 334, was primarily distributed at IV2016, V0605, V1713, and V2114, and it showed abundances of up to 0.8% (V1713, 150 m). Indeed, the surface water ecotype contributed to the total Nitrospinae abundance at this depth range. OTU 334, which belonged to Nitrospinaceae, showed 100% similarity with sequences from seawater of the Atlantic Ocean, the East China Sea, the interfaces of deep-sea brines, and the Red Sea. Within the intermediate water ecotype, OTU765 peaked at 300–1000 m, where it accounted for up to 0.47% of the total abundance (V312, 1000 m). OTU765 belonged to *Nitrospina* and displayed 100% similarity with a sequence from seawater from the Northeast Subarctic Pacific Ocean (1000 m deep). However, among the deep (bottom) water ecotypes, OTU 298 increased from 800 m to the bottom, reaching up to 0.3% at V1312 (5000 m). OTU 298, which belonged to Nitrospinaceae, displayed 99.75% similarity with sequences from the deep water of the Gulf of Mexico and abyssal seawater from the South Pacific Gyre. Thus, distinct nitrifier assemblages might adapt to divergent ecological niches and be responsible for efficient nitrification at different spatiotemporal regimes.

### 3.9. Quantitative PCR Analysis

As shown in Figure 7, the overall bacterial 16S rRNA gene copy numbers generally decreased northward. The bacterial 16S rRNA gene copy numbers were typically high at depths of above 100–200 m. The maximum levels, which ranged from 2.10 × 10^6^ to 2.24 × 10^7^ copies mL^−1^, overlapped with the maximum chlorophyll *a* levels and were generally observed at 30–100 m. A rapid decline by an order of magnitude was observed at 300 m (150 m at V0605, 800 m at IV1713); below which, the gene copy numbers decreased gradually with the depth to the lowest level of 10^3^–10^4^ copies mL^−1^ near the bottom of the water column.

We found that the overall Thaumarchaeotal 16S rRNA gene copy numbers tended to increase northward, reaching the maximum level at V1713; after which, they decreased to the lowest at V2114. The upper peak of the Thaumarchaeotal 16S rRNA copies was located at 150 and 200 m (6.6 × 10^4^–9.2 × 10^5^ copies mL^−1^), just below the peak of chlorophyll *a*, exhibiting the highest copy level at 200 m at V1312, followed by 150 m at V1713. Secondary peaks were generally distributed at 300 m and 500 m (6.6 × 10^4^–5.0 × 10^5^ copies mL^−1^), with the highest level observed at 500 m at V1312, followed by 800 m at V1713. The copies then tended to decrease with the depth to near the bottom (approximately 10^3^ copies mL^−1^). However, IV1713 and V0605 also showed additional peaks at 1000 m and 1250 m, respectively. Thaumarchaeotal *amoA* gene copies reached a maximum of 5.5 × 10^4^–1.2 × 10^6^ copies mL^−1^ at 150–200 m, corresponding to the upper peak of Thaumarchaeota, with the highest levels at V1312, followed by V1713 and IV2016. The *amoA* copies then declined with the depth to the lowest level of approximately 10^1^ copies mL^−1^ near the bottom. Similar depth distributions have also been repeatedly observed across wide-ranging locations, including in the Sargasso Sea [76], the Central Pacific [77], the Equatorial Pacific [78], and the Atlantic [79,80].

We observed that definite *Nitrospina* 16S rRNA gene copy numbers ranged from 200 to 1500 m in depth, with the highest copy numbers located at 200–1000 m (1.24 × 10^4^–1.30 × 10^5^ copies mL^−1^).

### 3.10. Correlation Analysis and Potential Mechanisms of the Nitrifying Process

The correlation analysis revealed a positive linear correlation between the mean content of chlorophyll *a* (10–150 m) and full-depth (10–4000 m) NO_3_^−^ (*R*^2^ = 0.798), and NO_3_^−^ at 200 m (*R*^2^ = 0.6142) and ≥300 m (300–4000 m; *R*^2^ = 0.7739) among the different stations (Figure 8a). This suggested that the deeper water could supply NO_3_^−^ to promote chlorophyll *a* production (organic matter) at the surface waters. These findings are consistent with those reported by Tyrrell [81], who found that NO_3_^−^ was a potentially important factor influencing primary productivity in the ocean. The correlation analysis also revealed that the mean Thaumarchaeotal 16S rRNA copies at 500–1500 m (*R*^2^ = 0.7279, Figure 8b) and across the water column (150–4000 m; *R*^2^ = 0.6714) showed a linear positive correlation with *Nitrospina* at 500–1500 m. Moreover, the full-depth Thaumarchaeotal 16S rRNA copies (*R*^2^ = 0.9007, Figure 8c) showed a linear positive correlation with nitrate (150–4000 m), while the *Nitrospina* 16S rRNA copies at 500–1500 m showed a linear positive correlation with nitrate at 500–1500 m (*R*^2^ = 0.6258) and the full depth (*R*^2^ = 0.6042). These findings indicated that Thaumarchaeota and Nitrospinae interdependently drove nitrate production across the water column. The correlation analysis also revealed that the full-depth Thaumarchaeotal 16S rRNA copies (*R*^2^ = 0.7748, Figure 8d) had a linear positive correlation with chlorophyll *a*. Therefore, Thaumarchaeota abundances might be partially determined by exported organic matter that could supply ammonia via remineralization, as previously reported [82]. Based on these findings, we speculated that the supply of nitrate via underlying water stimulated chlorophyll *a* production in surface waters, which enhanced the export of organic matter and ammonia substrate. As a consequence, Thaumarchaeota increased in abundance and, in turn, coupled with *Nitrospina*, produced and replenished nitrate, which enhanced primary production again. Hence, feedback between phytoplankton productivity (chlorophyll *a*) and nitrate concentration mediated with Thaumarchaeota and *Nitrospina* formed at large-scale regions.

It should be noted that, with the exception of V1713, the correlation studies demonstrated a linear negative correlation between the mean chlorophyll *a* content (10–150 m) and temperature (10–300 m) (*R*^2^ = 0.7055) (Figure 8e), as well as between NO_3_^−^ and the temperature at 10–300 m (*R*^2^ = 0.5952). Therefore, we proposed that low-temperature upwelling enhanced NO_3_^−^ transportation and proximally favored primary production. Additionally, a linear negative correlation was found between chlorophyll *a* and the DO minimum value (*R*^2^ = 0.7028) (Figure 8f), suggesting that high levels of O_2_ might be consumed below the highly productive surface waters. The correlation analysis also revealed that the mean content of NO_3_^−^ ≥ 300 m and DO ≥ 300 m (*R*^2^ = 0.6354) and the DO minimum values (*R*^2^ = 0.8788; Figure 8g) were negatively related. These findings indicated that NO_3_^−^ formation processes preferably occurred in low DO environments. These findings were consistent with the results of a previous study showing that the nitrification process occurred preferentially under low DO concentrations [16]. While the results reported by Peng et al. [16] were based on the vertical distribution at a specific environment, we further demonstrated that this relationship also existed among different stations in a large-scale study. We further found that, at 150–4000 m and the DO minimum depth, the Thaumarchaeotal 16S rRNA copies were negatively related to the DO minimum value (*R*^2^ = 0.5058; *R*^2^ = 0.715, Figure 8h), respectively, implying that Thaumarchaeota preferred areas with low-oxygen concentrations. Thus, high levels of phytoplankton increase organic matter export from underlying waters. The respiration processes then reduce the DO concentration, promoting high Thaumarchaeota abundance and favoring nitrate production. This regulation process replenishes nitrate consumed by phytoplankton and, consequently, maintains a balance. Hence, DO changes could also act as a regulator of chlorophyll *a*–nitrate feedback.

However, V1713 simultaneously displayed relatively high levels of chlorophyll *a*, nitrate, and DO. We speculated that deep and strong mixing (Appendix A) could transport nitrate upward from the deep layer, fueling chlorophyll *a* production. Below the mixing layer, a high level of Thaumarchaeota coupling to Nitrospinae might support nitrate formation under oxic environments. The Thaumarchaeota shallow water ecotype might partly contribute to nitrate formation at the euphotic zone. This would be consistent with the suggestion that deep water mixing is responsible for increases in Thaumarchaeota abundance in the southern oceans [71,83].

## 4. Conclusions

In the future, the ocean may experience major shifts in biogeochemical cycles triggered by the expansion and intensification of deoxygenation in response to climate change. This may also lead to the formation of feedback loops between oxygen concentrations, biological productivity, and nutrient cycling mediated by microbial activity to maintain ecosystem stability.

Here, the DO minimum values of 14.30 to 66.86 μmol L^−1^ along a ~12,000-km transect from 18° S to 13° N in the Tropical Pacific Ocean were observed at depths of 300 and 500 m. Although DO explained only 3.50% of the community variations, several OTUs associated with the nitrogen and sulfur cycles displayed significant negative correlations with DO, suggesting oxygen sensitivity of biochemical processes in the ocean. These species could act as reservoirs that might bloom as DO drops.

Our study also revealed natural feedback operated to maintain the nitrate balance and contribute to primary production. Specifically, nitrate is transported upward to phytoplankton for new C production in the energy-rich sunlit zone. Organic matter (chlorophyll *a*) is then exported downward. This favors Thaumarchaeota and *Nitrospina* growth, which, in turn, produces and replenishes nitrate that can be used for phytoplankton productivity. The downward movement of organic matter (chlorophyll *a*) might also enhance DO consumption, thereby enhancing Thaumarchaeota growth and nitrate formation.

## Figures and Tables

**Figure 1 microorganisms-11-02172-f001:**
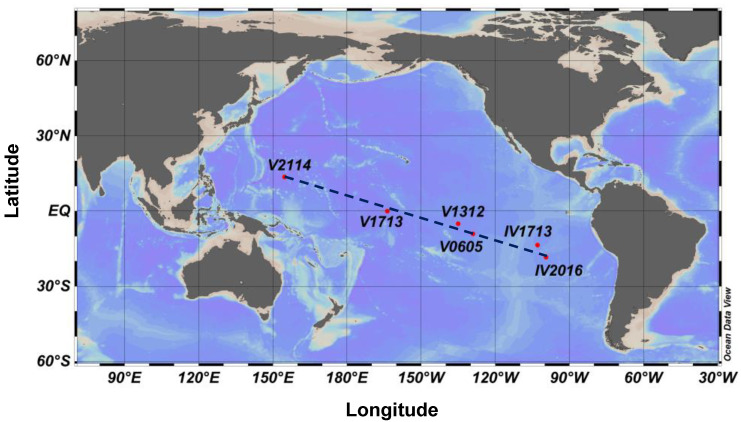
Location of the sampling stations along a transect in the Tropical Pacific Ocean.

**Figure 2 microorganisms-11-02172-f002:**
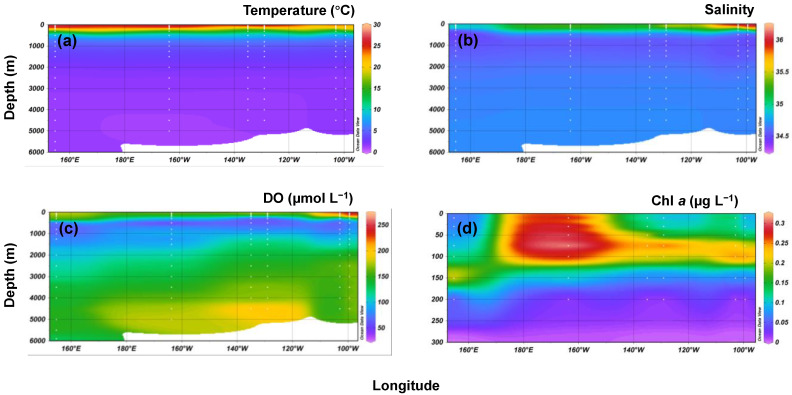
Temperature (**a**), salinity (**b**), dissolved oxygen (DO) (**c**), and chlorophyll *a* (Chl *a*) (**d**) along a transect in the Tropical Pacific Ocean.

**Figure 3 microorganisms-11-02172-f003:**
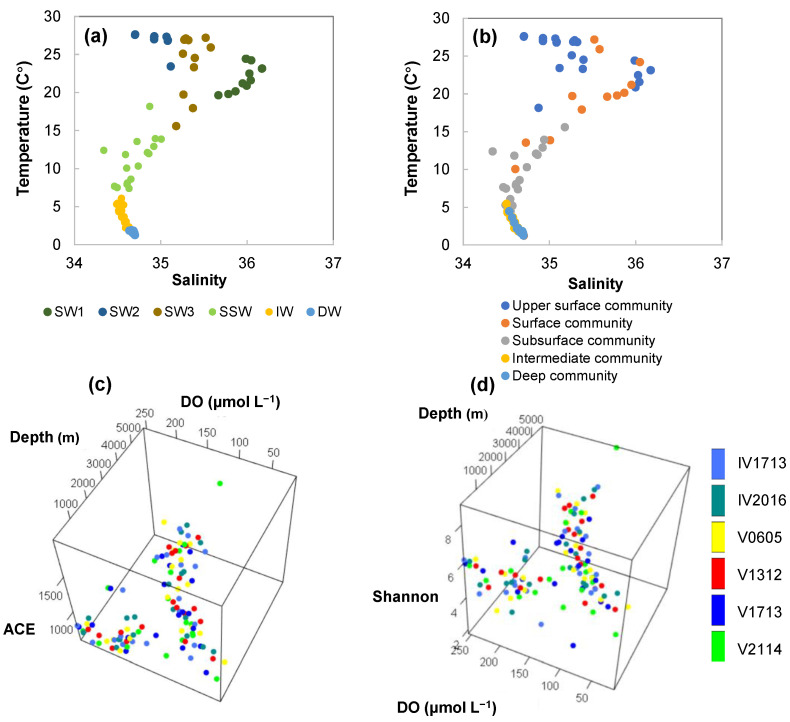
Water masses (**a**) and prokaryotic community structures (**b**) along temperature and salinity gradients. Alpha diversity indices, ACE Index, (**c**) and Shannon Index (**d**) along the depth and DO gradients.

**Figure 4 microorganisms-11-02172-f004:**
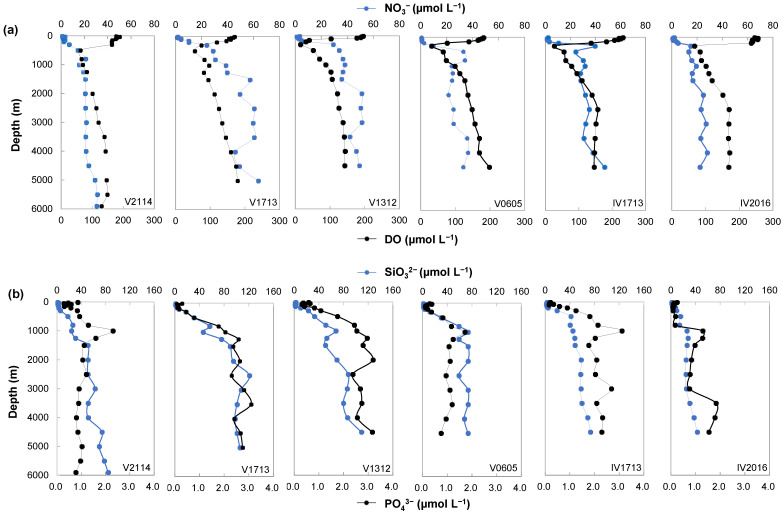
Depth profiles of physicochemical parameters, including dissolved oxygen (DO) and the NO_3_^−^ (**a**), SiO_3_^2−^, and PO_4_^3−^ (**b**) concentrations along the water columns.

**Figure 5 microorganisms-11-02172-f005:**
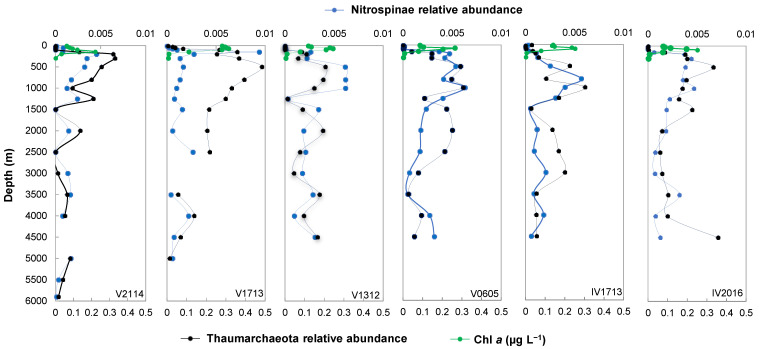
Depth profiles of chlorophyll *a* (Chl *a*) concentrations and the relative abundance of Thaumarchaeota and Nitropinae.

**Figure 6 microorganisms-11-02172-f006:**
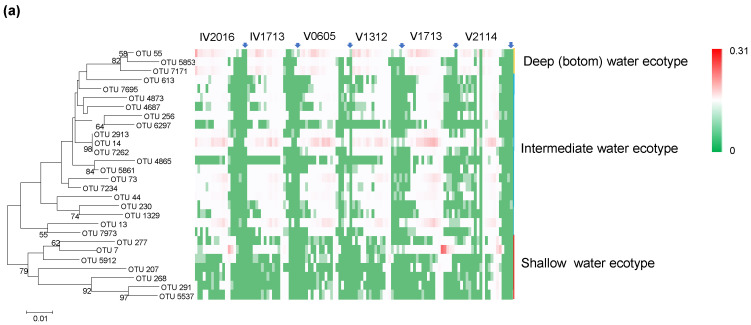
Phylogenetic trees and relative abundance heat maps of Thaumarchaeota (**a**) and Nitrospinae (**b**) OTUs identified throughout the water column of the samples collected in this study. A neighbor-joining tree was constructed from 16S rRNA gene amplicon sequences. A Bootstrap analysis was conducted with 1000 replicates to estimate support for the branches. The scale bars represent 1% and 2% sequence divergence, respectively. The blue arrows indicate a depth of 10 m at the different stations.

**Figure 7 microorganisms-11-02172-f007:**
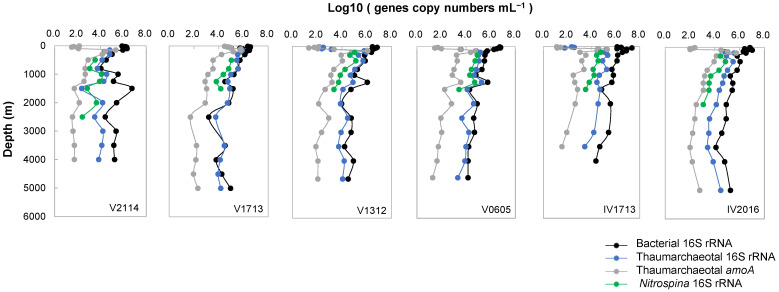
Horizontal and vertical distribution of log 10-transformed copy numbers of bacterial 16S rRNA, Thaumarchaeotal 16S rRNA and *amoA*, and *Nitrospina* 16S rRNA genes.

**Figure 8 microorganisms-11-02172-f008:**
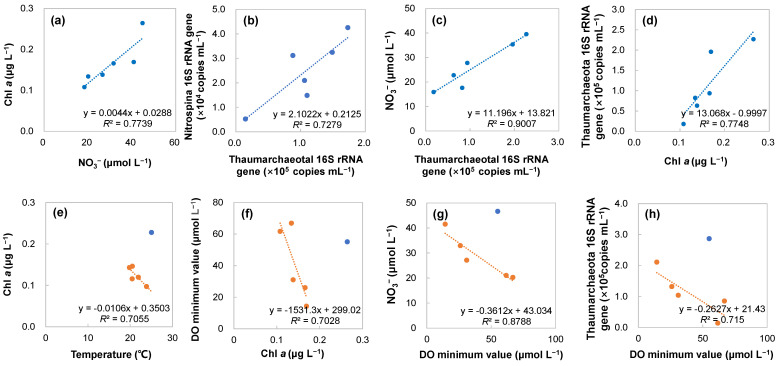
Correlation between NO_3_^−^ (≥300 m) and chlorophyll *a* (10–150 m) concentrations (**a**); Thaumarchaeotal and *Nitrospina* 16S rRNA gene copies (500–1500 m) (**b**); Thaumarchaeotal 16S rRNA gene copies and nitrate concentrations (150–4000 m) (**c**); chlorophyll *a* concentration and full-depth Thaumarchaeotal 16S rRNA gene copies. (**d**); temperature (10–300 m) and chlorophyll *a* concentration (**e**); chlorophyll *a* concentration and DO minimum value (**f**); NO_3_^−^ concentration (≥300 m) and DO minimum value (**g**); Thaumarchaeotal 16S rRNA gene copies and DO value at DO minimum depth (**h**) among different stations. The blue solid circles in (**e**–**h**) denote Station V1713.

**Table 1 microorganisms-11-02172-t001:** Primers used to quantify bacterial 16S rRNA, Thaumarchaeotal 16S rRNA and ammonia monooxygenase subunit A (*amoA*), and *Nitrospina* 16S rRNA genes.

Genes	Primers and Sequences (5′–3′)	Fragment Length (bp)	Annealing Temperature (°C)	Reference
Bacterial 16S rRNA gene	515f (GTGYCAGCMGCCGCGGTAA) 806r (GGACTACNVGGGTWTCTAAT)	291	55	[41]
Thaumarchaeotal 16S rRNA gene	TH95f (AATAAGGGGTGGGCA) TH279r (TACCGTCTACCTCTCCCACT)	184	53	[42]
*Nitrospina* 16S rRNA gene	NitSSU130f (GGGTGAGTAACACGTGAATAA) NitSSU282r (TCAGGCCGGCTAAMCA)	152	49	[43]
Thaumarchaeotal *amoA* gene	Arch-*amoA*f (CTGAYTGGGCYTGGACATC) Arch-*amoA*r (TTCTTCTTTGTTGCCCAGTA)	256	54	[44]

**Table 2 microorganisms-11-02172-t002:** Water masses, temperature, salinity, and depth ranges of the sampling stations.

Water Masses	SW1	SW2	SW3	SSW	IW	DW
Temperature (°C)	19.65–24.43	23.43–27.65	15.59–27.20	7.38–18.16	2.22–6.09	1.25–1.88
Salinity	35.67–36.17	34.70–35.11	35.18–35.58	34.34–35.01	34.49–34.64	34.64–34.70
Station	Depth (m)					
V2114	–	10–150	-	200–500	800–2000	≥2500
V1713	–	10	30–200	300–500	800–2000	≥2500
V1312	–	10–50	75 and 100	150–500	800–2000	≥2500
V0605	100 and 150	–	10–75	200–500	800–2000	≥2500
IV1713	10–100	–	150	200–500	800–2000	≥2500
IV2016	30–200	–	10	300–500	800–2000	≥2500

**Table 3 microorganisms-11-02172-t003:** Changes in OTUs with dissolved oxygen (*p* < 0.01; |*R*| ≥ 0.5).

Domain	Phylum (Class)	OTUs	Number of OTUs	Positive + Negative −
Archaea	Euryarchaeota	Thermoplasmata	4	−
		Marine Group III	1	−
	Thaumarchaeota	Nitrosopumilaceae	15	−
Bacteria	Marinimicrobia	Unclassified	6	−
	Acidobacteria	Unclassified	3	−
	Verrucomicrobia	Unclassified	4	−
		*Roseibacillus*	1	−
	Actinobacteria	Acidimicrobiia	3	−
		Solirubrobacterales	1	−
		*Corynebacterium lipophiloflavum*	1	−
	Bacteroidetes	Cytophagales	1	−
		Flavobacteriales	1	−
		Flavobacteriaceae	4	+
	Gemmatimonadetes	Unclassified	3	−
	Nitrospinae	*Nitrospina*	2	−
	Cyanobacteria	Unclassified	2	+
		Synechococcales	2	+
		*Phalacroma*	2	+
		*Aureococcus anophagefferens*	1	+
	Planctomycetes	Unclassified	2	−
	Alphaproteobacteria	Unclassified	6	−
		Unclassified	8	+
		Rhodobacteraceae	2	+
		Rhodospirillales	3	−
		Rickettsiales	5	+
		Sneathiellaceae	1	−
		*Candidatus* Pelagibacter ubique	3	+
		*Candidatus* Puniceispirillum	1	+
	Deltaproteobacteria	Unclassified	2	−
		SAR324	4	−
	Gammaproteobacteria	SAR86	2	−
		*Candidatus* Thioglobus	2	−
		*Prochlorococcus*	1	+
		Unclassified	2	+
		Unclassified	8	−
		Arenicellaceae	1	−
		*Pseudohongiella*	1	−
		Thiotrichaceae	1	+
		SUP05	3	−
		*Woeseia*	1	−

## Data Availability

Raw 16S rRNA sequence data generated in this study were deposited in the GenBank database under accession numbers SAMN30058985–SAMN30059100.

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
