# Peer review of "Prokaryote Distribution Patterns along a Dissolved Oxygen Gradient Section in the Tropical Pacific Ocean"

_microorganisms, 2023, doi:10.3390/microorganisms11092172_

Round 1

Reviewer 1 Report

I offer some suggestions on presentation that might help readers.

Prokaryote Distribution Patterns Along a Dissolved Oxygen 2
Gradient Section in the Tropical Pacific Ocean
Abstract:
OK

Figures & Tables (read firstly)
Figure 1: Panels b, c, d have Y axes scales of 0 to 6000 m, but panel e) has a scale from 0 to 300.
I suggest trying a log scale Y axes and put all panels on the same scale.

Table 1: fine
Table 2:  OK
Figure 2: OK
Figure 3:
I suggest turning off shadowing on symbols, it looks blurry (on my screen).

The axes tick labels are tightly spaces for NO3-, but widely spaced for DO and for PO43-.
I suggest moving Panel A & B a bit further apart, it was hard to figure out which labels go with which axes.
Figure 4:
Any message on the relation b/t chlorophyll and Thaumarchaeota is obscured b/c chlorophyll points fall almost on the axes.
Perhaps a log scale Y Axes?

Table 3
The boundary b/t Archaea and Bacteria is not properly spaced.  Acidobacteria downwards should be under 'Bacteria', but appear to be under 'Archaea'.
Why does the justification shift partway down the OTU column?
Figure 5
I work in this general area, and should be able to understand Figure + Legend + Caption as a unit.
But I have no idea what this Figure + Legend + Caption shows us;  once I read the paper I might figure it out, but Figure + Legend + Caption, as a unit,  should be comprehensible to an informed reader.

Figure 6
Figure 8: Multiple Y axes are not labelled on the figures, only mentioned in the Caption.
And some Y axes are labelled incorrectly, I think (8f)?
What is 'DO minimum value'?

Conclusions:
"Downward movement of organic matter (chlorophyll a) might also enhance DO consump- 603 tion, thereby controlling Thaumarchaeota growth and nitrate formation. "

Figure 8h shows highest levels of Thaumarchaeota rRNA gene at lowest DO minimum, so it looks like DO consumption stimulates Thaumarchaeata.

I read quickly in haste but English expression seems good.

Reviewer 2 Report

microorganisms-2520476

Prokaryote distribution patterns along a dissolved oxygen gradient section in the Tropical Pacific Ocean

General comments:

The present study presents relevant information about the prokaryotic distribution pattern coupled with physical and biogeochemical variables and dissolved oxygen gradients in the entire water column along a 7,000-km transect from 18°N to 13°S in the Tropical Pacific Ocean, using molecular analyses for the microbial community identification. The present work is extremely interesting as it gives a larger perspective of what is happening to the biogeochemical processes and it helps to understand predictive future oceanic nitrogen cycles under global climate change.

In general, I believe the manuscript is well written and structured. I have only some suggestions that are described in the specific comments below.

Specific comments:

Keywords: I suggest that you use different keywords, as they should be different than the already used in the title.

Materials and Methods

Figure 1:  I would use a larger map, to see the location of the sampling sites in a more global view of the pacific ocean. Also, I would not put already the results of the environmental parameters in the methods section. And you should mention what are the red arrows in the figure 1b means, and also the vertical lines in fig. 1e.

Page 3, Line 99: I would suggest a table with the specific depths that you sampled in each site. Maybe as supplementary material. You could also include data of the environmental parameters.

Page 4, Line 140: I would put this sub-section “2.4 Bioinformatics analysis and statistical analysis” at the end of the methods section.

Page 6, Line 212: When you refer to Table 2 in the beginning of the sentence, I do not think is the same Table 2 in the paper. Maybe is a typo, and is actually Table 1.

The quality of English language is fine.
